# Assessing Heat Management Practices in High Tunnels to Improve the Production of Romaine Lettuce

**Muzi Zheng [1,\*], Brian Leib [1], David Butler [2], Wesley Wright [1], Paul Ayers [1], Douglas Hayes [1] and Amir Haghverdi [3]**

[1] Department of Biosystems Engineering and Soil Science, University of Tennessee, 2506 E.J. Chapman Drive, Knoxville, TN 37996-4531, USA; bleib@utk.edu (B.L.); wright1@utk.edu (W.W. & P.A.); dhayes1@utk.edu (D.H.)

[2] Department of Plant Sciences, University of Tennessee, 2431 Joe Johnson Dr., Knoxville, TN 37996-4531, USA; dbutler@utk.edu

[3] Department of Environmental Sciences, University of California, Riverside, 900 University Avenue, Riverside, CA 92521, USA; amirh@ucr.edu

\* Correspondence: mzheng3@vols.utk.edu

**Abstract:** A three-year experiment evaluated the beneficial effects of independent and combined practices on thermal conditions inside high tunnels (HTs), and further investigated the temperature impacts on lettuce production. Specific practices included mulching (polyethylene and biodegradable plastic films, and vegetative), row covers, cover crops, and irrigation with collected rainwater or city water. The study conducted in eastern Tennessee was a randomized complete block split-split plot design (RCBD) with three HTs used as replicates to determine fall lettuce weight (g/plant) and lettuce survival (#/plot), and the changes in soil and air temperature. The black and clear plastic mulches worked best for increasing plant weight, but when compared to the bare ground, the higher soil temperature from the plastics may have caused a significant reduction in lettuce plants per plot. Moreover, the biodegradable mulch did not generate as much soil warming as black polyethylene, yet total lettuce marketable yield was statistically similar to that for the latter mulch treatment; while the white spunbond reduced plant weight when compared with black plastic. Also, row covers provided an increased nighttime air temperature that increased soil temperature, hence significantly increased lettuce production. Cover crops reduced lettuce yield, but increased soil temperatures. Additionally, irrigation using city water warmed the soil and provided more nutrients for increased lettuce production over rainwater irrigation.

**Keywords:** cover crop; lettuce production; irrigation; mulch; row cover; temperature variations

## 1. Introduction

High tunnels (HTs) are simple, plastic covered, greenhouse-like structures that do not utilize heaters or ventilation fans. Even without heaters or fans, HTs allow producers to lengthen the growing season and protect plants from extreme weather conditions (e.g., hail, frost, or strong wind), hence increasing the profitability and sustainability of organic farms [1]. Compared to crop production in open fields, other advantages of HTs highlighted by Lamont et al. [2] also include: (1) preventing rainfall from wetting and splashing soil onto fruit and foliage, thus creating cleaner products with less disease; (2) increasing water-use efficiency; (3) improving crop environmental conditions; and (4) reducing soil compaction and insect invasion. Additionally, the benefits of less electricity consumption, lower startup costs, and fewer maintenance efforts make HT systems more competitive than standard greenhouses [3].

High-valued crops, such as lettuce, tomatoes, and other leafy greens, produced inside HT systems can also compensate for the increased cost of HT systems when compared to open field production. Studies also show that the microclimates inside HTs can be modified so that the growing season can be lengthen from 1 to 4 weeks in spring and 2 to 8 weeks in fall [4]. Therefore, HTs are currently being adopted by small- and mid-scale producers in order to take advantages of market seasonality with higher profits [5,6].

However, there are several limitations for the sustainability of HT systems. First, since HTs block rainfall, ground/well or treated water needs to be used for crop irrigation. As well, lettuce is considered moderately sensitive to irrigation salinity, since the salts in soil and water might cause the retardation of crop growth, reduction of lettuce head formation, or even the necrotic lesions on the leaf margins [7]. To combat this sensitivity, rainwater can be collected and reserved in black polyethylene tanks, and then used inside HTs through drip irrigation. A rain harvesting system (RHS) is considered beneficial in regions where treated water is inadequate and costly, and helps in reducing local flooding in some low-lying areas. This study investigates additional benefits of RHS when black polyethylene tanks are used; black tanks can capture solar energy and warm up enclosed rainwater, and this warmer water can warm the soil during drip irrigation.

Secondly, the poor insulation properties of the plastic-covered HT structure do not provide significant heat retention at night under cold climates. Without a favorable growth microclimate, crop performances inside HTs might be inhibited or even terminated. Accordingly, surface mulch and row covers are practices that could help retain heat while providing other advantages, such as reducing weed competition, alleviating soil crusting, modifying the radiation budget of the soil surface around the plants, and conserving water by inhibiting evaporation from the soil surface [8]. A two-year lettuce experiment in the open field found that colored polyethylene mulches (clear, white, and black) significantly increased the overall lettuce production by 33%, 40%, and 39%, respectively, compared to bare ground [9]. Additionally, for other high-value crops, a study showed that black polyethylene mulch combined with an HT system significantly increased overall pepper production in regard to crop height and leaf area, while in the same system, clear plastic mulch significantly raised ground temperature, thus decreasing the number of days to the first flower when compared with black polyethylene mulch and bare ground [10]. As well, biodegradable mulch can eliminate plastic disposal issues, including the labor cost of disposal [11,12]. There is limited research on how biodegradable mulches within HT systems affect soil temperature and crop productivity. But in open field tests, outdoor use of biodegradable mulches had no significant effect on overall pepper yield and quality in Australia; although, it raised soil temperature slightly compared with paper mulch [13]. A study from Moreno [14] also indicated that biodegradable mulches did not significantly increase tomato production in Central Spain, but Miles et al. [15] observed improved tomato production using biodegradable mulches when compared with bare ground in northwestern Washington, USA. Since lettuce production could be limited from weed competition and seedling establishment problems, appropriate polyethylene and biodegradable mulches might reduce the costs of weed management, while minimizing root damage, also retaining favorable soil temperatures for lettuce growth and development [16]. Similar to surface mulches, floating row covers are considered as additional thermal protection for crops from wind and frost inside a HT. There have been many studies showing that the combination of mulching and row covers under field conditions significantly increased soil and air temperatures for favorable crop growth conditions [17]. However, some studies indicated that although HT air temperature was slightly higher than outside, lettuce development was still slow, since the air temperature around the plant's full height was not optimal [18]. Therefore, more investigation on the impacts of row cover and colored plastic/biodegradable mulches under HTs are needed to maximize heat retention.

The last limitation is related to the decomposition of soil organic matter due to continuous cropping in HTs, which may negatively influence nutrient/water holding capacity in soil. The common solution of adding organic supplements to the soil requires energy at each step as long as the supplements need to

be collected and delivered to HTs. Accordingly, an effective practice is to pre-plant a legume cover crop which has the ability to fix nitrogen, supply crop nutrients and improve soil structure in HT systems. This study investigated incorporating cover crop residues into the ground and leaving the residues on the soil surface (vegetative mulch). Teasdale and Daughtry [19] found that cover-crops have positive effects on weed control and Liebman and Davis [20] observed reduced maximum-soil-temperature under vegetative mulches in the daytime, along with the benefit of increased summer production. Even though cover crops provide many benefits, there is limited research on the effects of cover crop residues incorporated into the ground in combination with polyethylene mulches covering the soil surface, and their effects on thermal protection in HT systems.

Overall, this study aims to provide recommendations to small- and mid-scale producers on how to improve thermal protection with low input sustainable practices to improve lettuce production in HTs. Rain water, surface mulch, row covers, and cover crops have many benefits, but this study will focus on the benefits to heat management. Zheng et al. [21]'s study showed how thermal energy conservation benefited spring pepper production, while this study aims to assess the yield performance of fall Romaine lettuce using the same thermal energy conservation practices.

## 2. Material and Methods

### 2.1. The Gothic Type High Tunnel

Three experimental high tunnels were used in the experiment, that were orientated N–S, and located at the University of Tennessee's Organic Crop Unit in Knoxville, TN in the United States (latitude 35.88° N, longitude 83.93° W, and elevation 252 m). These HTs have peaked roofs, vertical side walls and the dimension of the structures are 9 m in width and 15 m in length (Figure 1). Each end door covers an opening which is 2.45 m tall and 3.35 m wide, and side curtains run the entire length of the HT. The end doors were opened almost every day to provide natural ventilation while the side-curtains were only opened during warm weather when more air was need for proper ventilation. In cold periods, doors and side curtains were closed at night to preserve thermal energy, and only the end doors were opened in the daytime to reduce accumulated heat and humidity inside HTs. Rainwater was collected and stored in black polyethylene tanks and was delivered using gravity pressure and solar power. Rainwater storage tanks in the first HT were elevated with cinder blocks to provide gravity pressure for irrigation with drip tubing, while the other two HTs utilized a solar powered pump to deliver rainwater via drip tapes with 10 psi of pressure [21].

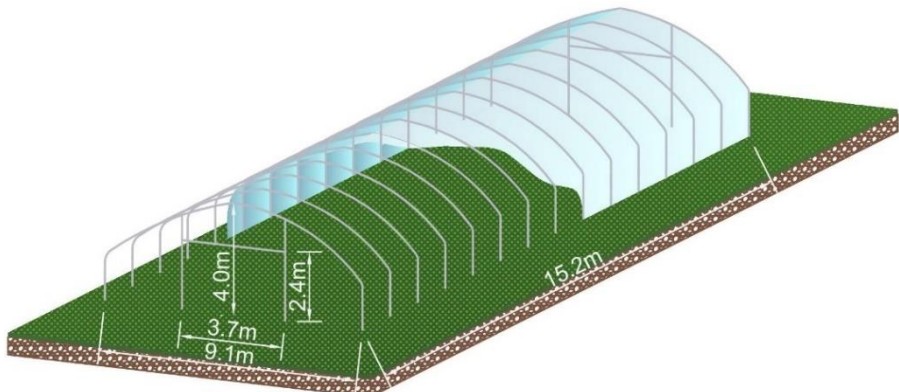

**Figure 1.** Isometric view of a high tunnel with the covering cut away for clarity) [21].

Romaine Lettuce ('coastal star') was grown during the fall seasons of 2011, 2013, and 2014. Specifically, five different surface mulches with a bare soil control, and two sources of irrigation water were applied in fall 2011. Thus, there were six beds in each HT, and these six different treatments were randomly arranged in different beds. Figure 2a shows that mulch treatment was laid out in six

rows, including: (1) white biodegradable spunbond (spun), (2) vegetative mulch from a leguminous cover crop (veg), (3) bare ground (bare), (4) black polyethylene film (black), (5) biodegradable brown paper (paper), and (6) black biodegradable biobag (biobag). The dimension of each bed was 12.19 m in length and 0.91 m in width with double rows of lettuce planted in each bed. Double planting rows were placed 0.3 m apart and 0.46 m apart between lettuce transplants. There were 48 plants per row and rows were divided into four plots. Lettuce was transplanted on 30 September 2011 in half of each house (snapdragon flowers were added to the other half of the HT for a different research project), and then harvested on 7 December 2011. Next, in the fall of 2013, there were five surface mulches and a bare control, with or without a row cover. Figure 2b shows that the mulch distribution was: (1) black polyethylene film (black), (2) vegetative mulch from a cover crop (veg), (3) bare ground (bare), (4) black polyethylene film with a cover crop (blackCC), (5) clear polyethylene film with a cover crop (clearCC), and (6) clear polyethylene film (clear). Once inside temperature fell below approximately 7 °C, row covers were applied to half of the crop during the night. Lettuce was transplanted on 2 October 2013 and harvested on 4 December 2013. Finally, in the fall of 2014, a total of 12 treatments were used, including two water resources (rainwater and city water), two surface mulches (black and blackCC) and a bare ground control, with or without a row cover (Figure 2c). Lettuce was transplanted in each half of the HT on 29 September 2014 and harvested on 5 December 2014.

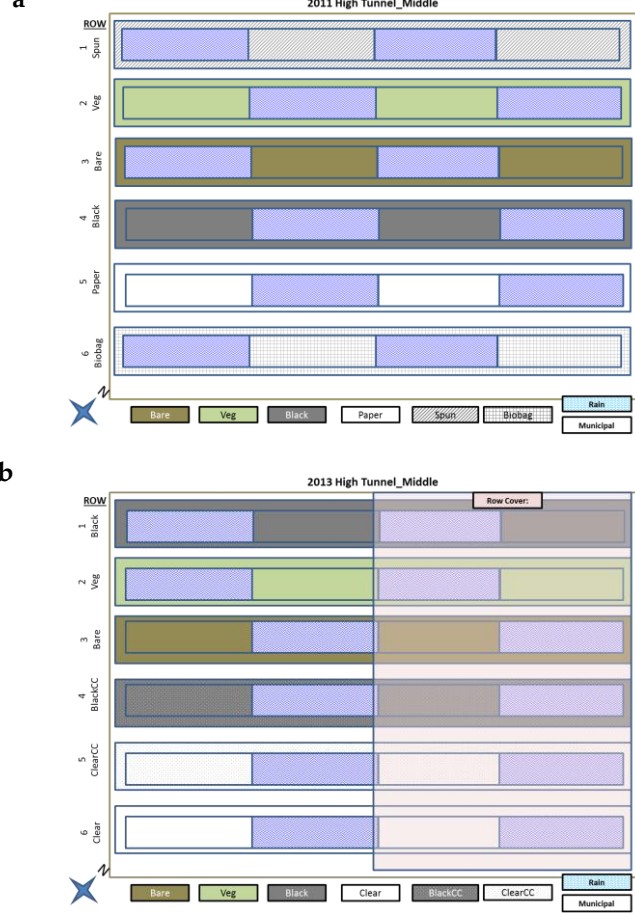

**Figure 2.** *Cont.*

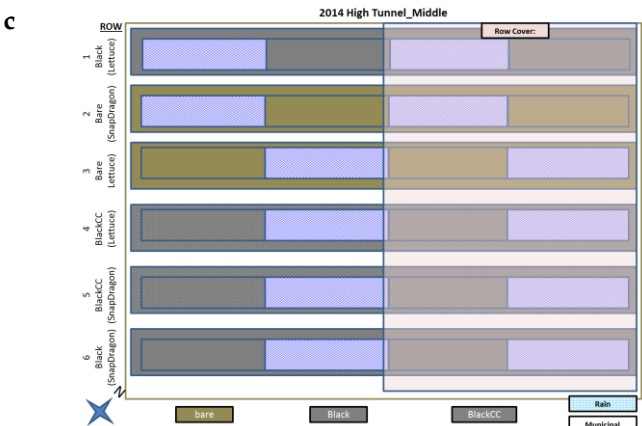

**Figure 2.** The lettuce production of high tunnels (HTs) in 2011, 2013, and 2014 followed a split-split-plot design.

To determine statistical differences, the experiment was a randomized complete block design (RCBD), based on a split-split-plot sub-design for lettuce production and temperature variations. In addition, soil at the experimental site was a Dewey silt loam. Lettuce yield was determined by plant weight (g/plant) and plants per plot (#/plot). Climatic analysis were divided into two time periods, including early fall (EF) from planting through October and late fall (LF) from November to early December, and statistical analyses of lettuce yield values and temperatures variations were performed by SAS statistical software (SAS Institute Inc., Cary, NC, USA).

*2.2. Climatic Monitoring and Instrumentation*

Two meteorological stations were used to monitor conditions with one placed outside and another placed inside the high tunnels [21]. The outside station was 6.0 m away from the middle HT and set to 4.0 m above the ground. The inside station was located at 1.5 m above the ground at the center of the middle experimental HT. The sensors used for each station included a pyranometer for solar radiation, a cup anemometer and wind vane for wind speed and direction, along with a capacitive chip and platinum resistance thermometer, for relative humidity and air temperature, respectively. Additionally, there was a total of 15 thermistors applied to measure inside air temperature at 1.2 m above the ground surface. An array of 12 thermistors to monitor air temperatures at the canopy level were located 30 cm above the ground surface. Half of these thermistors were under the row cover and half were outside the row cover. Moreover, soil temperature was measured in each combination for the mulch, row cover, and water using 24 thermistors placed 10 cm into the ground. Additionally, the water temperatures were measured at the bottom of polyethylene storage tanks and in the supply line of city water. All the thermistors described above were potted in solar reflecting white epoxy and all the sensors were measured every 15 s and recorded every hour by a CR1000 datalogger (Campbell Scientific, Inc.: Logan, Utah, USA). The specifications for the sensors are listed in Table 1.

**Table 1.** Environmental parameters measured during the experiments [21].

| Measured Parameters | Sensors | Range and Accuracy | Location |
|---|---|---|---|
| Solar radiation $Q_{solar}$ (Wm$^{-2}$) | LI200X Pyranometer (LI-COR.INC) | 0 to 3000 Wm$^{-2}$ with ±5% | Inside and outside weather station |
| Air temperature $T_i$ (K) Relative humidity $H_R$ (%) | HMP60 Probe (Vaisala.INC) | −40 to 60 °C with ±0.6% ±3% RH over 0 to 90%, ±5% RH over 90 to 100% | Inside and outside weather station |
| Wind speed $U$ (ms$^{-1}$) Wind direction (°) | R.M.Young Wind Sentry (Campbell Sci. Inc) | 0 to 50ms$^{-1}$ with ±1% 0 to 360° with ±5° | Outside weather station |
| Air temperature (°C) | Thermistor (PS104J2) [†] | ±0.1 °C with 0.026 °C | An array of 15 thermistors all over the inside of middle HT located 1.5 m above the ground |
| Canopy temperature (°C) | Thermistor (PS104J2) [†] | ±0.1 °C with 0.026 °C | 12 thermistors located 30 cm above the ground between crop rows |
| Soil temperature (°C) | Thermistor (PS104J2) [†] | ±0.1 °C with 0.026 °C | 24 thermistors located 10.16 cm below the ground |
| Water temperature (°C) | Thermistor (PS104J2) [†] | ±0.1 °C with 0.026 °C | Polyethylene storage tank bottom for rainwater and supply line of city water |

[†] means the thermistors are made in our lab, which are non-linear sensors following a polynomial response to temperature that is defined by the Steinhart–Hart equation.

## 3. Results and Discussion

The HT affected inside microclimate in terms of solar radiation, air and soil temperature, and relative humidity and wind speed. There were 1380 observations of weather data measured over three falls of 2011, 2013, and 2014. Generally, 9% of total solar radiation was reflected by the HT's polyethylene covering (dataset in 2011 was disregarded due to sensor failure). During the EF (early fall) with the end-doors and side-curtains fully opened, air temperatures inside the HT were consistently higher by 0.5–6.0 °C compared to outside; while in LF (late fall) with the side curtains rolled down, the inside maximum air temperature during the day was from 1 to 11 °C higher when compared to the outdoor conditions. Moreover, outside and inside thermistor readings indicated that night time air temperatures inside the HT were 0.6–2.0 °C higher than outdoor temperatures, and the average nighttime soil temperature inside the HT was 1–3 °C higher than outside over the span of the three falls [21].

Statistically, mulch treatments were considered the main contributing factor to the lettuce productivity in the falls of 2011, 2013, and 2014 ($p < 0.001$) (Figure 3). In 2013 and 2014, black consistently produced a higher yield than bare, with an increase of 71% (2013) and 37% (2014) in g/plant. However, no significant difference was found in lettuce yield between using black and bare plots in 2011. Clear mulch provided the maximum production in g/plant in 2013 but with no significant difference compared to the black mulch treatment. In addition, Table 2 shows that the increased crop yield using black was related to its soil-warming ability when compared with bare. Although some temperature values were missing due to sensor failures in LF 2011, black had a positive impact on soil temperature, which was constantly higher than the bare treatment by 0.8–1.3 °C (2011), 0.6–0.9 °C (2013), and 0.9–1.3 °C (2014). The soil temperature rise under black mulch can promote faster lettuce growth and development, although the lettuce survival (#/plot) with black was reduced approximately 20% when compared to bare in 2013. The soil-warming ability underneath clear was even greater than that under black, by as much as 1.0 °C, but the overall lettuce yield did not improve significantly in spring 2013. After transplanting in EF, soil temperatures were warmer by 1.4 °C (day) and 2.1 °C (night) for clear, and 0.6 °C and 0.9 °C for black when compared to the bare. The higher measured soil temperatures under the mulches do not adequately reflect, the extremely hot air emitted from the transplant holes or the high temperature of the mulch itself before the crop canopy can shade the

plastic mulches. These higher temperatures seem beneficial to lettuce growth (lettuce weights per plant) but they may not be beneficial to lettuce survival at transplant, since the plants per plot were significantly reduced in clear (10%) and black (20%) plots compared with the bare.

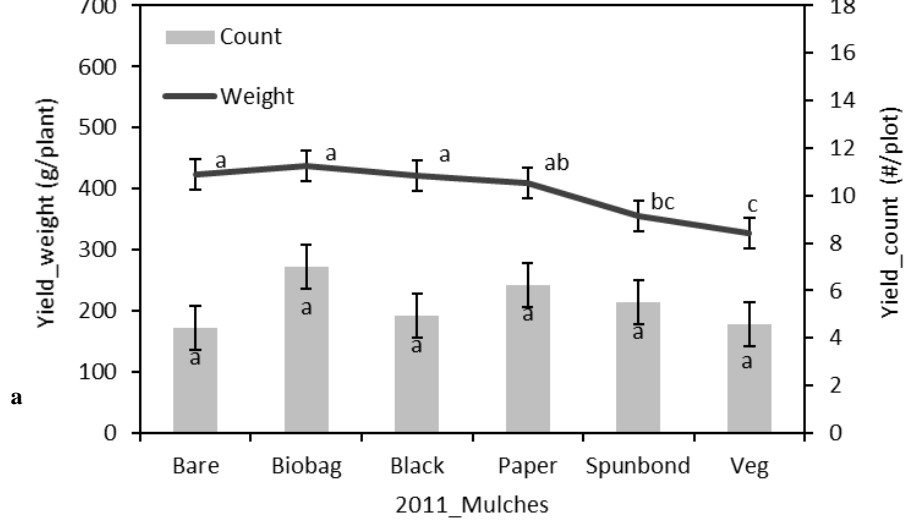

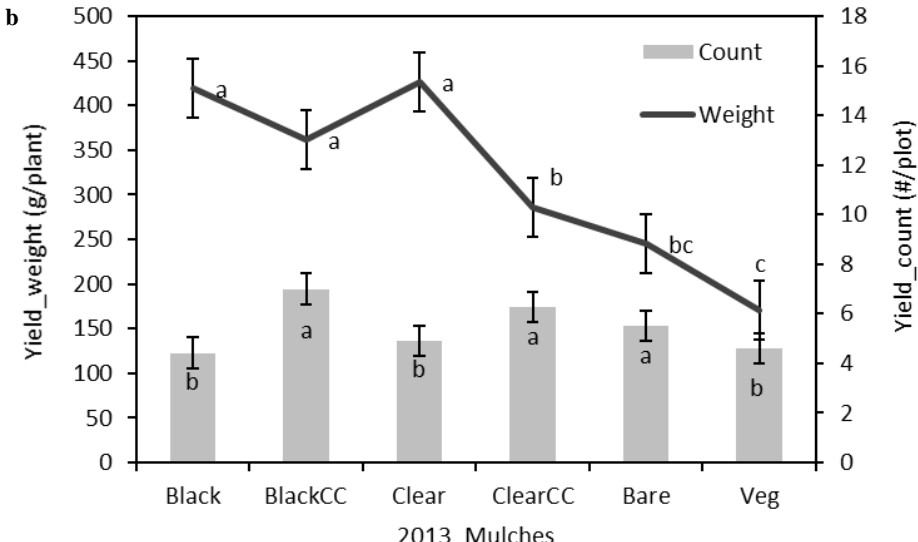

**Figure 3.** *Cont.*

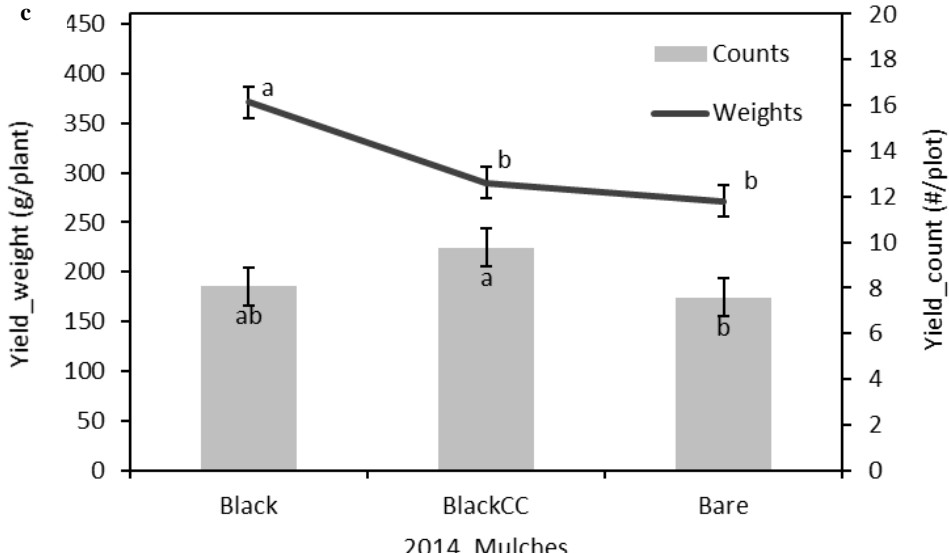

**Figure 3.** Marketable lettuce weight per plant and lettuce numbers per plot under the effects of different mulch over the falls of the years 2011 (**a**), 2013 (**b**), and 2014 (**c**) in HTs. Values followed by the same letter are not significant different according to Fisher's LSD (*p* = 0.1).

**Table 2.** Marketable lettuce weight per plant and lettuce numbers per plot, and average soil temperatures at 10 cm depth under different mulch treatments during the day and night in the early and late falls of 2011, 2013, and 2014.

| Year | Mulch | Lettuce Yield | | Soil Temp_Day | | Soil Temp_Night | |
|---|---|---|---|---|---|---|---|
| | | **Weight** | **Count** | **EF[†]** | **LF[†]** | **EF[†]** | **LF[†]** |
| | | (g/plant) | (#/plot) | (°C) | | (°C) | |
| | Black | 421.6 a | 7.7 a | 18.22 a | - | 14.42 a | 15.39 a |
| | Biobag | 436.9 a | 8.3 a | 17.12 b | - | 13.26 c | 14.17 c |
| 2011 | paper | 409.1 ab | 8.7 a | 16.68 bc | - | - | - |
| | Spunbond | 355.7 bc | 9.3 a | 16.21 c | - | 13.01 d | 13.99 d |
| | Bare | 424.2 a | 9.2 a | 17.06 b | - | 13.59 b | 14.13 cd |
| | Veg | 327.5 c | 9.2 a | 17.06 b | - | 12.96 d | 13.52 d |
| | Sig.level | *0.0311* | *0.73* | *0.0115* | - | *<0.0001* | *<0.0001* |
| | Black | 419.4 a | 4.4 b | 19.97 c | 14.12 b | 19.74 c | 14.31 e |
| | BlackCC | 361.4 a | 7.0 a | 20.08 c | 14.18 b | 20.52 b | 14.66 b |
| | Clear | 425.8 a | 4.9 b | 20.80 b | 14.20 b | 20.86 b | 14.66 b |
| 2013 | ClearCC | 285.6 b | 6.3 a | 21.47 a | 14.69 a | 21.81 a | 15.02 a |
| | Bare | 244.9 bc | 5.5 a | 19.41 d | 13.26 d | 18.82 d | 13.43 e |
| | Veg | 170.4 c | 4.6 b | 19.22 e | 13.01 d | 18.36 d | 13.28 e |
| | Sig.level | *<0.0001* | *<0.0001* | *<0.0001* | *<0.0001* | *<0.0001* | *<0.0001* |
| | Black | 371.1 a | 8.1 ab | 18.38 a | 13.07 a | 19.48 a | 13.86 a |
| 2014 | BlackCC | 290.0 b | 9.8 a | 18.24 b | 12.97 a | 19.32 a | 13.76 a |
| | Bare | 271.2 b | 7.6 b | 17.30 c | 12.12 b | 18.16 b | 12.86 b |
| | Sig.level | *<0.0001* | *0.1765* | *<0.0001* | *<0.0001* | *<0.0001* | *<0.0001* |

\* EF and LF were characterized by diurnal soil temperature in specific early and late spring. Temperature values are daily means from 10:00 a.m. to 16:00 p.m. and nightly means from 21:00 p.m. to 8:00 a.m. Values followed by the same letter are not significantly different according to Fisher's LSD (*p* = 0.1).

Moreover, the application of the dark colored biobag and paper were not significantly different when compared to black for lettuce yield in 2011. However, white spunbond produced less marketable lettuce weight per plant by 13% than paper, but the difference in #/plot was not at a significant level. All of the degradable mulch plots provided significantly lower soil temperatures than black by 1.0–1.2 °C (biobag), 1.4–2.0 °C (spunbond), and approximately 1.6 °C (paper). The greater reduction of

soil temperatures produced by spunbond (1.4–2.0 °C) compared to those of biobag and paper mulch may be the reason for significantly decreased lettuce production under spunbond when compared to black.

In additional, polyethylene mulch with cover crops (blackCC and clearCC) had the potential to save more lettuce heads in each plot by 21%–60% (#/plot). This may be because more air was able to penetrate through the cover crop residues, thus the thermal stress from the plastic mulch surface was able to be mitigated when the film edges contacted the lettuce transplants in the warm EF period. But the plastic mulch with cover crops significantly reduced marketable lettuce weight per plant (18% in blackCC and 33% in clearCC), when compared to non-cover crop treatments as an average for 2013 and 2014. Specifically, the average soil temperatures under blackCC in EF and LF of 2013 were around 0.3–0.8 °C warmer compared with black at night, whereas the temperature differences between clearCC and clear were 0.5–0.7 °C and 0.3–0.9 °C warmer during the day and night, respectively. Thus, although the cover crops had a higher soil-warming ability and saved more lettuce in each plot, limited nutrient availability caused by immobilization of nitrogen during the decomposition of cover crop residues may have inhibited the growth of each lettuce head, as the incorporated cover crop and residual soil nutrients served as the only fertilizer source. In comparison, veg always significantly provided the lowest lettuce production. It appears that the vegetative mulch shielded the transfer of solar energy to the soil and cooled the soil to a point where lettuce development was detrimentally delayed. Results confirmed that veg had a significant decrease of soil temperature (0.6 °C in 2011 and 0.3 °C in 2013) than that in bare. Therefore, veg treatment with the coolest soil temperature was working against a goal of making better use of solar energy to extend the growing season.

For the LF, additional thermal protection was provided by row covers for lettuce growth (Figure 4). Row cover played a significant role in increasing g/plant for 2014 ($p < 0.0001$). The lettuce weight per plant utilizing a row cover was improved approximately 36% as an average of all treatments (black, blackCC, and bare) when compared to all treatments without row covers. Moreover, positive interactions between the applications of mulches and row covers were found on yield ($p < 0.1$) in fall 2013. The black mulch with row cover provided the largest lettuce production and the row covers increased lettuce weight by 28% over black mulch alone. While some of the lower yielding plots had greater lettuce weight when a row cover was applied, bare and veg yielded significantly more lettuce weight, approximately 45% and 73% more lettuce weight per plant respectively, under row cover. Moreover, blackCC, clear, and clearCC with a row cover had the potential to improve the total marketable production (g/plant) compared with those without using row covers in fall 2013, but the improvement was not to a significant level. In addition, the row cover can significantly increase canopy and soil temperature by 0.8–2.3 °C and 0.1–0.9 °C at night, respectively. Although soil temperature increases from row covers interacted differently with the mulch treatments—0.4 °C in black, 0.6 °C in blackCC, 0.1 °C in clear, 0.8 °C in clearCC, 0.7 °C in bare, and 0.2 °C in veg—the row covers had a positive effect on lettuce production in all plots.

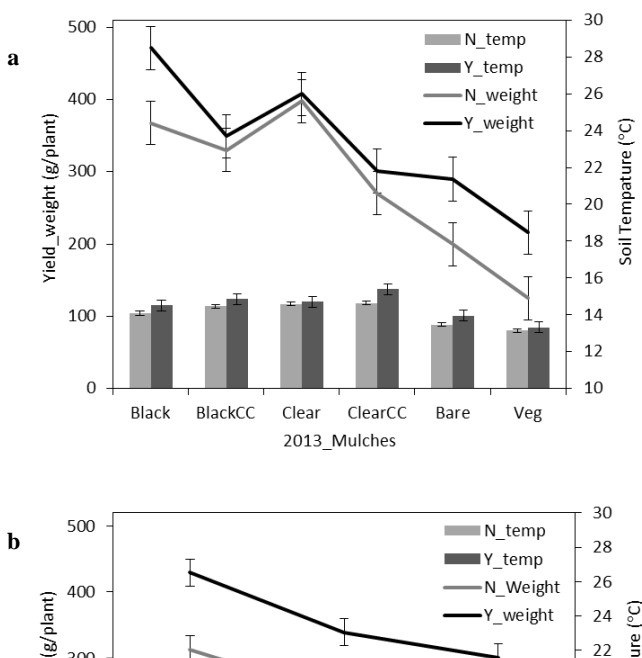

**Figure 4.** Interactions of mulch and row cover on lettuce yield (weight/plant), and soil temperature (°C) inside high tunnels during the late fall (LF) of 2013 and 2014. '*Y*' represents the mulch with a row cover and '*N*' means the mulch without a row cover.

Finally, lettuce yield was improved by using city water compared to rainwater applied with drip irrigation (Figure 5). As an average of all mulches, city water potentially increased the total lettuce weight per plant by 16% in 2011, 12% in 2013, and 8% in 2014, and also increased total lettuce numbers per plot by 13% in 2013 (Table 3). Although the numeric values of lettuce number per plot were potentially higher by using city water than rainwater, the differences were not at a significant level. In 2011, the biobag mulch and bare irrigated using city water significantly improved lettuce weights by 23% and 36%, respectively, compared to rainwater ($p < 0.0001$), although in subsequent years, city water did not significantly increase yield in the bare soil treatment. In 2013, black and veg mulch yield was significantly increased when city water was used for irrigation by 36% and 20% in lettuce weight per plant, respectively; and in 2014, black mulch with city water also significantly improved the lettuce production approximately by 17% (g/plant) over rainwater. The differences of lettuce yield using two water types were assumed to be explained by the impacts of water temperature on soil temperature. Figure 6 shows that the rainwater temperature from the solar tank was consistently cooler than that in gravity tanks. Additionally, in the EF of 2011, 2013, and 2014, the average temperature of rainwater stored in the solar tank was significantly higher than the city water in all years by 2.4 °C; but during the LF periods, the rainwater from solar tanks was significantly cooler than the city water by 2.2 °C in 2011, 1.3 °C in 2013 and 0.8 °C in 2014. Although water temperatures changed inconsistently between the EF and LF, soil temperatures responded in a similar manner over different mulches. Table 3 shows that city water generally produced warmer soil temperatures than rainwater during the entire fall seasons, with several exceptions in 2011 (biobag, spunbond, and bare), but all these variations cannot change the fact of greater lettuce production in city water plots (23% for biobag, 13% for spunbond, and 35% for

bare). In summary, the temperature of city water was cooler than rainwater in EF and warmer during LF. However, the overall effect was for city water to warm the soil on average during each time period. As early fall progresses into late fall, rain water would be expected to cool in relation to city water, because the above ground tanks are exposed to less daily solar radiation and longer cooler nights, while soil temperature around the city water pipes lags behind the above ground heat loss of the rain water tank; i.e., soil temperature decline lags air temperature decline in the fall. The comparison of water temperature data does follow this expected pattern. However, the contradiction between EF water temperature and soil temperature differences is less easily explained. Perhaps the rainwater temperature was not representative of the water delivered for irrigation due to the sensor's location in the tank and temperature gradients within the tank caused by tank surface heating and cooling. In addition, the temperature transition from warmer rainwater to warmer city water would not occur precisely at the interface of EF to LF. There was most likely an intermixing of more and less heat transfer from the irrigation sources to the soil, in combination with air temperature's effects on soil temperature. Overall. the response of different mulches to irrigation water type shows that lettuce yield was improved by applying city water when compared to rainwater, and warmer soil temperatures appear to be an important factor where city water was applied. Another reason for increased lettuce production using city water may be related to water quality. Periodic water samples were analyzed and revealed the pH of city water was consistently around 6.4, while the rain water changed over time, but mainly was around at 6.7 to 7. Additionally, city water had more nutrients than rainwater: around 11 to 16 times higher levels of K, Mg, and Ca which may help promote lettuce growth and development.

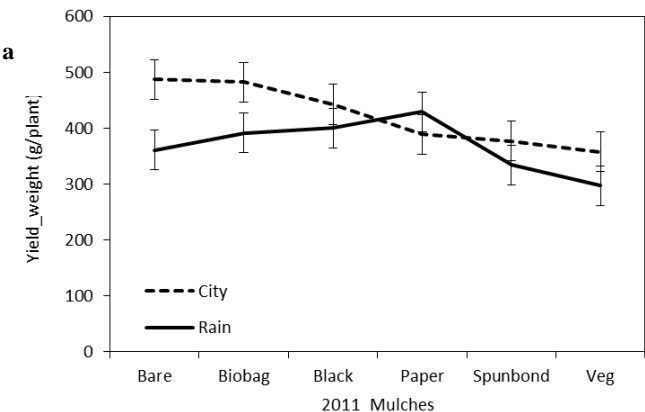

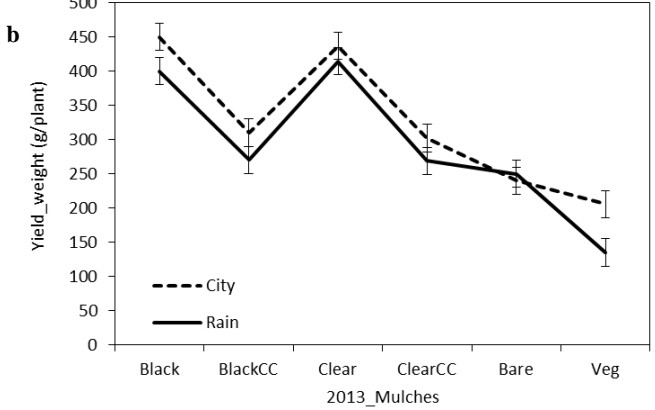

**Figure 5.** *Cont.*

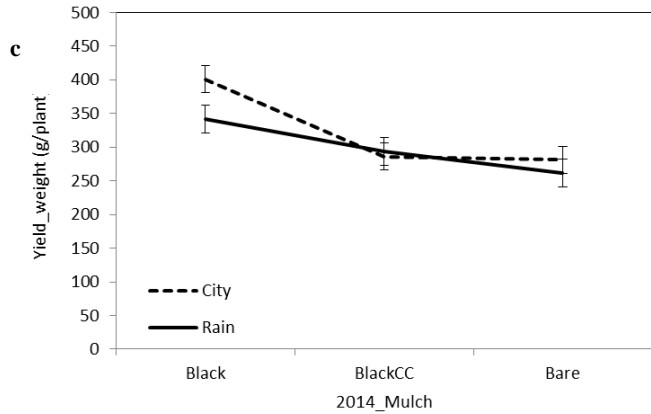

**Figure 5.** Marketable weight of lettuce (mean ± SE) under the effects of different irrigation water (rain and city) over three fall seasons of the years 2011 (**a**), 2013 (**b**), and 2014 (**c**) in high tunnels.

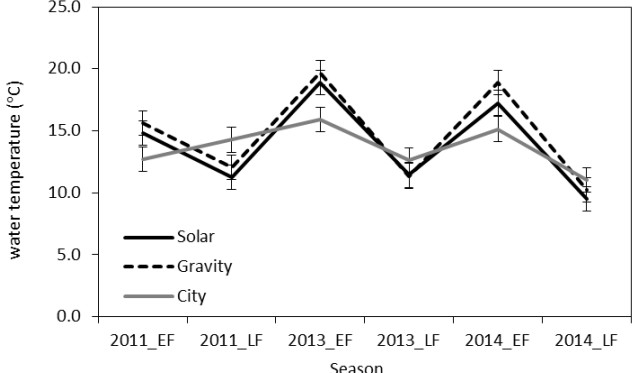

**Figure 6.** Fluctuations in average water temperature (mean ± SE) for three fall seasons of years 2011, 2013, and 2014. Gravity and Solar represent the rainwater inside the gravity and solar tanks, respectively, and City represents the municipal water from the underground pipes.

**Table 3.** Interactions of mulch and water on the lettuce yield and soil temperature inside high tunnels during all three falls of years 2011, 2013, and 2014.

| Year | Mulch | Water | Lettuce Yield | | Soil Temp_Day | | Soil Temp_Night | |
|---|---|---|---|---|---|---|---|---|
| | | | Weight | Count | EF[†] | LF[†] | EF[†] | LF[†] |
| | | | (g/plant) | (#/plot) | (°C) | | (°C) | |
| 2011 | Black | City | 442.67 abc | 10.00 a | 18.37 a | - | 14.44 a | 15.45 a |
| | | Rain | 400.58 bcd | 7.33 ab | 18.07 ab | - | 14.40 a | 15.34 a |
| | Biobag | City | 482.52 ab | 8.67 ab | 16.94 bcd | - | 13.10 fg | 13.83 e |
| | | Rain | 391.33 cd | 8.00 ab | 17.30 abc | - | 13.42 def | 14.52 bc |
| | Paper | City | 388.67 cd | 9.67 ab | 16.80 bcd | - | 13.48 cde | 14.53 bc |
| | | Rain | 429.50 abc | 9.00 ab | 16.56 c | - | - | - |
| | Spunbond | City | 377.33 cde | 8.33 ab | 15.90 d | - | 12.87 g | 13.77 e |
| | | Rain | 334.00 de | 10.00 a | 16.53 c | - | 13.15 efg | 14.21 d |
| | Bare | City | 487.33 a | 8.33 ab | 17.11 bc | - | 13.19 efg | 13.79 e |
| | | Rain | 361.04 cde | 7.0 b | 17.01 bc | - | 13.99 b | 14.48 c |
| | Veg | City | 357.69 cde | 9.00 ab | 17.34 abc | - | 13.61 cd | 14.72 b |
| | | Rain | 297.33 e | 9.00 ab | 16.78 bcd | - | 13.78 bc | 14.55 bc |
| | Mulch | | *0.0311* | *0.73* | *0.0115* | - | *<0.0001* | *<0.0001* |
| | Water | | *0.0158* | *0.38* | *0.8917* | - | *0.0018* | *<0.0001* |
| | Mulch × Water | | *0.3177* | *0.59* | *0.5621* | - | *0.0678* | *<0.0001* |

Table 3. *Cont.*

| Year | Mulch | Water | Lettuce Yield | | Soil Temp_Day | | Soil Temp_Night | |
|---|---|---|---|---|---|---|---|---|
| | | | Weight | Count | EF[†] | LF[†] | EF[†] | LF[†] |
| | | | (g/plant) | (#/plot) | (°C) | | (°C) | |
| 2013 | Black | City | 450.0 a | 5.17 bcd | 20.15 c | 14.28 c | 20.04 cd | 14.57 d |
| | | Rain | 400.0 a | 3.67 d | 19.79 d | 13.97 d | 19.43 ef | 14.04 e |
| | BlackCC | City | 310.3 abc | 8.00 a | 20.09 c | 14.16 c | 20.48 c | 14.64 cd |
| | | Rain | 270.0 ab | 6.00 cd | 20.07 c | 14.20 c | 20.57 c | 14.68 cd |
| | Clear | City | 437.0 a | 5.00 bcd | 20.83 b | 14.24 c | 21.23 b | 14.74 c |
| | | Rain | 414.7 a | 4.8 bc | 20.77 b | 14.16 c | 20.49 c | 14.57 d |
| | ClearCC | City | 302.2 bcd | 6.17 bc | 21.54 a | 14.80 a | 21.78 ab | 15.10 a |
| | | Rain | 269.0 cd | 6.33 bc | 21.40 a | 14.59 b | 21.83 a | 14.93 b |
| | Bare | City | 239.7 de | 6.50 ab | 19.46 ef | 13.48 e | 18.97 fg | 13.57 g |
| | | Rain | 250.1 cd | 6.50 ab | 19.35 f | 13.03 g | 18.68 g | 13.29 h |
| | Veg | City | 205.6 de | 4.83 cd | 19.63 de | 13.86 d | 19.62 de | 13.88 f |
| | | Rain | 135.0 e | 4.33 d | 18.81 g | 13.28 f | 19.33 ef | 13.80 f |
| | Mulch | | *<0.0001* | *<0.0001* | *<0.0001* | *<0.0001* | *<0.0001* | *<0.0001* |
| | Water | | *0.3362* | *0.0576* | *0.0002* | *<0.0001* | *0.1511* | *<0.0001* |
| | Mulch × Water | | *0.0718* | *0.4984* | *0.005* | *<0.0001* | *0.2181* | *0.0023* |
| 2014 | Black | City | 400.9 a | 8.83 ab | 18.51 a | 13.17 a | 19.62 a | 13.98 a |
| | | Rain | 341.3 b | 7.33 ab | 18.26 b | 12.96 b | 19.34 b | 13.74 b |
| | BlackCC | City | 286.2 c | 10.17 a | 18.24 b | 12.97 b | 19.36 b | 13.78 b |
| | | Rain | 293.8 c | 9.33 ab | 18.24 b | 12.98 b | 19.27 b | 13.74 b |
| | Bare | City | 281.0 c | 6.00 b | 17.49 c | 12.25 c | 18.18 c | 12.89 c |
| | | Rain | 261.4 c | 5.6 ab | 17.11 d | 11.98 d | 18.13 c | 12.82 c |
| | Mulch | | *<0.0001* | *0.1765* | *<0.0001* | *<0.0001* | *<0.0001* | *<0.0001* |
| | Water | | *0.12* | *0.9542* | *0.0021* | *0.0072* | *0.0954* | *0.0192* |
| | Mulch × Water | | *0.1956* | *0.2697* | *0.0587* | *0.0893* | *0.4201* | *0.217* |

For values within mulch, water followed by the same letter are not significantly different according to Fisher's LSD (*p* = 0.1). EF[†] and LF[†] were characterized by nighttime soil temperature in specific EF and LF.

## 4. Conclusions

The three-years of data demonstrated that high tunnels combined with the application of surface mulches and the row cover, along with different irrigation water can be expected to have significant impacts on soil temperature, which influenced lettuce growth and development. The mulches had significant impacts on soil temperature, which was related to total lettuce yield over the falls of 2011, 2013, and 2014. Clear and black plastics all have good soil-warming ability, thus produced the highest lettuce weights. However, the thermal stress with the higher soil temperatures produced in clear and black plastics at transplanting may have caused a reduction of plants per plot when compared to the bare. The biodegradable biobag, spunbond and paper mulches did not heat the soil as much as the clear or black mulch, but the marketable lettuce yield produced by biobag and paper mulches were not statistically different compared with the black polyethylene mulch. In addition, the cover crops incorporated into the soil underneath the black and clear mulch had the potential to save more lettuces in each plot, but overall lettuce weight per head was reduced significantly when compared to the non-cover crop treatments. The reduction in yield from the cover crop may be due to less available nitrogen for plant use, as the organic matter is being decomposed. Moreover, vegetative mulch produced the coolest soil temperatures, and consistently generated the lowest lettuce yield, so is not recommended to be applied in HTs. Additionally, during cold nights, row covers can add protection in the LF, thus produced higher lettuce production when compared to groups without row covers. Finally, the temperature of city water was generally higher than rainwater except in the EF, and it could increase the soil temperature by 0.2–0.8 °C compared to the rainwater plots. Although there were several exceptions when soil temperatures were warmer using rainwater than city water, overall lettuce production was still increased by using city water. Additionally, city water generally provided

more nutrients than rainwater in terms of K, Mg, and Ca which may promote lettuce growth and development. Therefore, city water irrigation may not have just warmed the soil but also provided more nutrients for increased lettuce production.

**Author Contributions:** Conceptualization, M.Z. and B.L.; methodology, M.Z., B.L. and D.B.; software, M.Z. and B.L.; validation, M.Z. and B.L.; formal analysis, M.Z.; investigation, M.Z.; resources, B.L.; Data curation, M.Z., B.L., W.W., D.B., A.H.; writing—original draft preparation, M.Z.; writing—review and editing, M.Z., B.L., D.H., P.A., and D.H.; visualization, M.Z.; supervision, B.L.; funding acquisition, B.L.

**Funding:** This project was funded in part by Conservation Innovation Grants, NRCS, USDA (2013-2016) and Ag Research & Extension Innovation Grants (2011).

**Conflicts of Interest:** The authors declare no conflict of interest. The funders had no role in the design of the study; in the collection, analyses, or interpretation of data; in the writing of the manuscript, or in the decision to publish the results.

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
