# Peer review of "Assessing Heat Management Practices in High Tunnels to Improve the Production of Romaine Lettuce"

_agriculture, doi:10.3390/agriculture9090203_

Round 1

Reviewer 1 Report

Thank you for accepting the comments and do the corrections suggested in your article.

Author Response

Thanks fo your review comments. 

Reviewer 2 Report

Please see the comments in the attached reviewed manuscript.

Author Response

Please see the revision in the attached updated manuscript. Thanks for your review comments. 

This manuscript is a resubmission of an earlier submission. The following is a list of the peer review reports and author responses from that submission.

Round 1

Reviewer 1 Report

The current study was conducted to demonstrate the effect of various factors such as temperature and irrigation and mulching strategies on leafy vegetable production grown under high tunnel conditions. The study was conducted properly and the authors clearly show significant and interesting effects.

Author Response

We have already improved the english language, thanks for your comments.

Reviewer 2 Report

All corrections and comments are inserted in text. Please consult Instructions for Authors and Article Text  Template.

Abstract

This section is too long without some specific conclusion or recommendation for the application in practice.

Materials and Methods

This is very complicated research with many treatments and their interactions through three years. I am aware it is hard to explain it but try to be as simple as you can because it became confusional for reader.

Results and Discussion

Some important results as air temperature are not shown. At the other side, weight and number of plants per plot are shown both in figures and tables.Please change it.

Conclusions

This section is too long and too general. After the first sentence, emphasize and distiguish the best results with a recommendation for use in HT during the fall growing season.

Paragraph Author Contributions is missing (see Template)

Author Response

Point 1: Abstract is to long (maximum 200 words). Conclusion or recommendation is missing! are not numbered.

Response 1: The abstract length has already adjusted. Please check the updated manuscript.

Point 2: Are you sure about that mentioned temperature increasing under mulch caused plant number reduction? How do you explain that higher soil temperature under clear caused lower plant number reduction compared to black?

Response 2: The sensors for daily soil temperature were placed around 4 inches under the ground, thus the highest temperature was up to 21°C. But sometimes if the air temperature was quite high with improper ventilation, the surface temperature of clear plastics could be increased very quickly and may accidently burn many crop transplants. This observation was also found in the same author’s paper with fall pepper production in high tunnels. Therefore, the thermal stress increased under clear might be the possibility that we could guess to cause plant number reduction.

Point 3 & Point 4: This is already shown in Figure 4. There is no need to show data both in figure and in table. You may delete Figure 4 and add SD for weight and count into Table 2. These data are shown in Table 3 just without SE. Do not show datas in both ways.

Response 3 & Response 4: The data comparison between different mulch treatment would be more obvious using graphs in Figures. Tables are more like to see the temperature changes and its impact on lettuce yield.

Point 5: This section is too long and too general. After the first sentence, emphasize and distinguish the best results with a recommendation for use in HT during the fall growing season.

Response 5: Already adjusted in the manuscript.

Reviewer 3 Report

            The manuscript did a long-term experiments of mulch effects on the growth of lettuce in high tunnels. However, it is nothing new in this topic, there are a bunch of research reports on mulch topic. The major defect of the experiment design is that all the experiments were carried out in the early fall or late fall. The air temperatures were not presented, however, from the data of table 2 and table 3, the soil temperature-day was around 17-20°C and soil temperature-night was around 13-15 °C. These temperatures are almost the optimal for lettuce growth. The different mulches were definitely affected the soil temperatures, but only 0-2 °C. This minor temperature change will not affect the plant growth significantly when the whole temperature ranges are still within the optimal ranges. It will be much better if the experiments can carry out in the winter, when the soil or air temperatures drop down to the critical point, then the 2 °C increase by mulch will definitely improve the crop growth. So, my suggestion is that it is much better if the experiments were carried out in colder early spring, winter, to research the warming effects of mulch, or in the summer to research the cooling effects, instead of the best growing season.

          And the other important thing is that the mulch not only affect temperature. Mulch also has other advantages, such keep the soil moisture, inhibit weed. From my opinion, reducing evaporation and keeping the soil moisture are possibly the critical reasons to improve the lettuce growth, as the temperature was always in the optimal ranges during the experiment. However, this manuscript did not mention even one word on the soil moisture. And the manuscript also did not give the data of air temperature, which is also important.

Other suggestions:

    Figure 1 & 2, It is not necessary to put such kind figure here, as most if not all of the readers know the structures of high tunnels. And there are many different sizes and structures of HTs, not only this specific one. If you are using this specific one for your experiment, then you should move it to material and method part (and combine these two figures in to one).

    Line 76-78, What is the temperature raised? If you cited from others’ work, then put a citation.

    Line 158-159, “Also, rainwater was applied to the vegetable crops by two methods gravity pressure and solar power.” The methods should be presented in detail.

    Line 180-181, You are researching on lettuce, so why grow snapdragon flowers together? What is the purpose?

    Line 205 & Figure 4, All the two replicates of the treatments are in the same row. It is not a randomized complete block design. And missing some icons in figure 4.

    Line 259, What kind of relationship? such as positive or negative? linear or quadratic?

    Line 310, You did not measure the nitrogen, how do you know “reducing the nitrogen”.

    Line 349-351, Most of the data are not statistically significant in the table 3. The “water” did not show significant difference on yield.

    Line 379-381, You did not measure the K, Mg, Ca either. If these data are from other’s work, then cite it.

    Line 409-411, Day soil temperature of 21 C is not stress to most of plants, including lettuce.

    Line 418-419, Again, you did not measure the nitrogen contents either in soil nor in plant tissue. No data can support this conclusion.

    Line 423-424, Again, from table 3, the “water” did not show significant difference on yield. It is not correct conclusion.

    Line 427-428, again, you did not have any data to support this.

And the data of lettuce yield were used duplicated both in the figure and table.

    Generally, I cannot recommend this paper for publication at this status. For more detailed comments, please see the corrected manuscript.

Author Response

Point 1: Figure 1 & 2, It is not necessary to put such kind figure here, as most if not all of the readers know the structures of high tunnels. And there are many different sizes and structures of HTs, not only this specific one. If you are using this specific one for your experiment, then you should move it to material and method part (and combine these two figures in to one).

Response 1:  the adjustment has already been made in the updated manuscript.

Point 2: Line 76-78, What is the temperature raised? If you cited from others’ work, then put a citation.

Original Sentence: Compare to the ground or treated water, the warmer rainwater is transferred periodically to the soil through drip irrigation hence warmer soil temperature in a significant way for better HT production, especially in cold climates.

Response 2:  

This study investigates additional benefits of RHS when black polyethylene tanks are used; black tanks can capture solar energy and warm up enclosed rainwater and this warmer water can warm the soil during drip irrigation.

Point 3: Line 158-159, “Also, rainwater was applied to the vegetable crops by two methods gravity pressure and solar power.” The methods should be presented in detail.

Response 3:  

the adjustment has already been made in the updated manuscript as following: “Rainwater was collected and stored in black polyethylene tanks and was delivered using gravity pressure and solar power. Rainwater storage tanks in the first HT were elevated with cinder blocks to provide gravity pressure for irrigation with drip tubing, while the other two HTs utilized a solar powered pump to deliver rainwater via drip tapes with 10 psi of pressure (Zheng et al., 2019).”

Point 4: Line 180-181, You are researching on lettuce, so why grow snapdragon flowers together? What is the purpose?

Response 4:

Usually bell pepper and tomatoes are grown for some research purposes in the spring, then the crops are switched to head lettuce and snapdragon cut flowers in the fall. Lettuce are planted in half HT, snapdragon cut flowers were planted for other research purposes thus we won’t focus on this crop.

Point 5: Line 205 & Figure 4, All the two replicates of the treatments are in the same row. It is not a randomized complete block design. And missing some icons in figure 4.

Response 5:

It would be more accurate to say “Randomized complete block design with replicated treatments”

We didn’t find any missing icon; which icons are you referring to?

Point 6: Line 259, What kind of relationship? such as positive or negative? linear or quadratic?

Original sentences: Compared with bare, the increased crop yield using black was related to its soil-warming ability.

Response 6:

The relationship is positive, increased crop yield using black is due to the higher temperature under black mulch when compared to bare ground. Average soil temperatures under black were consistently greater than that in bare by 0.8-1.3°C (2011), 0.6-0.9°C (2013) and 0.9-1.3°C (2014).

Point 7Line 310, You did not measure the nitrogen, how do you know “reducing the nitrogen”.

Original sentences: A positive effect of incorporating cover crop residue under plastic mulch was higher soil temperature which could increase micro-organism activities that are breaking down the organic matter may be reducing nitrogen that is available to the plant (except the blackCC in 2014).

Response 7:

Pre-planting legume cover crop usually has the ability to fix nitrogen, supply crop nutrients and improve soil structure in HT systems. But in this study, cover crops allowed for greater heat transfer to soil and increased soil temperatures but lead to reduced overall lettuce yield. Thus, “reducing nitrogen” is considered as an assumption to explain why we think the cover crop has negative impact on lettuce production.

Point 8Line 349-351, Most of the data are not statistically significant in the table 3. The “water” did not show significant difference on yield.

Original sentences:

As an average of all mulches, city water increased the total lettuce weight per plant by 16% in 2011, 12% in 2013 and 8% in 2014, and also significantly increased total lettuce numbers per plot by 13% in 2013 (Table 3).

Response 8: the adjustment has already been made in the updated manuscript: “As an average of all mulches, city water potentially increased the total lettuce weight per plant by 16% in 2011, 12% in 2013 and 8% in 2014, and also increased total lettuce numbers per plot by 13% in 2013 (Table 3). Although the numeric values of lettuce number per plot were potentially higher by using city water than rainwater, the differences were not at a significant level.”

Point 9Line 379-381, You did not measure the K, Mg, Ca either. If these data are from other’s work, then cite it.

Response 9: we measured the nutrients in lab to get the results of 11-16 times higher levels of K, Mg, Ca in city water when compared to rainwater.

Point 10Line 409-411, Day soil temperature of 21 C is not stress to most of plants, including lettuce.

Original sentences: the thermal stress with higher soil temperature produced in clear and black plastics may cause reduction on plant numbers per plot when compare to the bare ground.

Response 10: the sensors for daytime soil temperature were placed around 4 inches under the ground, thus the highest temperature was up to 21°C. But sometimes if the air temperature was quite high with unproper ventilation, the surface temperature of clear or black plastics may be increased very quickly and may accidently burned many crop transplants. This observation was also found in the same author’s another paper with fall pepper production in high tunnels.

Point 11Line 418-419, Again, you did not measure the nitrogen contents either in soil nor in plant tissue. No data can support this conclusion.

Response 11: same with point 7.

Point 12Line 423-424, Again, from table 3, the “water” did not show significant difference on yield. It is not correct conclusion.

Response 12: same with point 8.

Point 13Line 427-428, again, you did not have any data to support this.

Response 13: same with point 9.
